# Immunoinformatics analysis of candidate proteins for controlling bovine paratuberculosis

**Maryam Sadat Moezzi**[1], **Abdollah Derakhshandeh**[1]*, **Farhid Hemmatzadeh**[2]

1 Department of Pathobiology, School of Veterinary Medicine, Shiraz University, Shiraz, Iran, 2 School of Animal and Veterinary Sciences, The University of Adelaide, Adelaide, South Australia, Australia

* drkhshnd77@gmail.com

## Abstract

### Background

Paratuberculosis is debilitating chronic enteritis usually characterized by diarrhea, decreased milk production, and progressive cachexia. *Mycobacterium avium* subspecies *paratuberculosis* (MAP) causes significant economic losses by affecting dairy herds globally. Development of protective vaccines is considered as one of the most effective controlling measures for MAP infections. In the current study, hydrophilic parts of MAP2191 and FAP-P proteins as two vaccine candidates were analyzed using immunoinformatics approaches.

### Methods

After selecting the most hydrophilic parts of MAP2191 and FAP-P, helper and cytotoxic T-cell epitopes of ht-MAP2191 and ht-FAP-P were identified. The immunogenic, toxicity and physicochemical properties were assessed. Secondary structures of these proteins were predicted, and their tertiary structures were modeled, refined, and validated. Linear and conformational epitopes of corresponding B-cells were recognized. Then ht-MAP2191 and ht-FAP-P epitopes were employed for molecular docking simulations.

### Results

The results indicated that ht-MAP2191 and ht-FAP-P were immunogenic, non-allergenic, and non-toxic and possess potent T-cell and B-cell epitopes. Eventually, these protein constructs were docked favorably against TLR4.

### Conclusion

According to the findings, ht-MAP2191 and ht-FAP-P could be effective protein-based vaccine candidates for paratuberculosis. It should be noted that to examine their efficacy, further in vitro and in vivo experiments are underway.

**Data Availability Statement:** All relevant data are within the article and its Supporting Information files.

**Funding:** This work was financially supported partially by a grant from Shiraz University (98GCU2M163973) and also Iran National Science Foundation (INSF) (No. 98023138). The funders had no role in study design, data collection and analysis, decision to publish, or preparation of the manuscript.

**Competing interests:** The authors have declared that no competing interests exist.

## 1. Introduction

*Mycobacterium avium* subspecies *paratuberculosis* (MAP) is an etiologic agent of paratuberculosis (PTB), or Johne's disease, leading to chronic infectious granulomatous enteritis in ruminants [1]. Young animals like calves become infected by ingesting contaminated colostrum, milk, or feces, and usually, the infection remains subclinical for about 2–5 years. The symptoms include diarrhea, progressive weight loss, and decreased milk production. Affected animals usually die after suffering cachexia and dehydration [2, 3]. A long subclinical phase with probable intermittent shedding before the high shedding clinical phase is among the significant characteristics of this disease in cows [4]. MAP infection results in notable economic losses in the dairy industry worldwide [5]. Due to similarities between PTB and Crohn's disease, it seems MAP might have a causative role in Crohn's disease in humans [6]. It has been found that MAP in the blood and intestine presence in Crohn's disease patients was seven times higher than in patients with any other bowel inflammation. [7, 8]. Because of direct and indirect economic costs, destructive effects on animal welfare, and increasing public health concerns, finding a way to control MAP infections is essential [9, 10]. Evaluating immunogenic antigens to develop efficient diagnostic tests and vaccine design is one of the solutions [11].

Mammalian cell entry (Mce) proteins as virulence-related proteins are functionally analogous to ABC transporters, and it appears they have roles in the lipid uptake system [12–14]. Although many studies discuss their function in the virulence of mycobacteria [15], the information about the virulence role of these conserved cell-wall proteins in MAP is limited. There are eight separate *mce* genes in the MAP K-10 reference genome, and *Mce*5 operon possesses a cluster of six homologs of the *mce*-family (*MAP2189* to *MAP2194*) genes. The genes of this operon are involved in MAP invasion, survival, and virulence processes [16, 17].

It has been shown that attachment and internalization of *M. avium subsp. paratuberculosis* through epithelial cells depends on the interaction between fibronectin attachment proteins (FAPs) and fibronectin [18, 19]. The FAPs are a family of fibronectin-binding glycoproteins expressed by some species of mycobacteria. They induce Th1 polarization and IFN-γ production in vitro [20].

In the last years, immunoinformatics tools and reverse vaccinology approaches have been applied to detect the antigenic regions of proteins of interest and design multi-epitope-based and subunit vaccines against cancers and pathogens [21, 22]. Compared to laboratory-based techniques, in-silico analysis is cost-effective, accurate, rapid, and reliable [23, 24].

This study aims to investigate the efficacy of mce2191 and FAP-P proteins for use in vaccine design against MAP and assess their ability to stimulate the immune system through immunoinformatics tools and servers. In this regard, highly antigenic epitopes (cytotoxic T-lymphocyte (CTL), helper T lymphocyte (HTL), and B-cell epitopes) were identified. Moreover, different immunological and physicochemical parameters were checked, including allergenicity, toxicity, and solubility. Then 3D models of the selected sequences were built, refined, and quality assessed. Finally, modeled structures were docked against TLR4 to ensure their efficacy in inducing an immune response.

## 2. Methods

### 2.1. Retrieval protein sequences

The amino acid sequences of MAP2191 and FAP-P were retrieved from the NCBI database. The conserved regions were determined, and the most hydrophilic parts were chosen through the ClustalX v2.1 program [25] and BioEdit v7.2 Software, respectively. The homology of the final sequences was inspected against the bovine proteome with the Protein BLAST server.

## 2.2. Antigenicity, allergenicity, toxicity, and solubility analysis

The antigenicity of extracted sequences was predicted by the VaxiJen v2.0 web server using the auto cross-covariance (ACC) transformation method. The web-based server AllergenFP v1.0 was also employed to find the allergenic profiles using a descriptor-based fingerprint approach [26, 27]. Toxicity properties and solubility tendency were estimated by the ToxinPred server and the Protein-Sol online server, respectively [28, 29].

## 2.3 Prediction of Cytotoxic T Lymphocyte (CTL) epitopes

Cytotoxic T lymphocyte cells play a significant role against intracellular pathogens and limit infections [30]. The NetMHC v4.0 serve with high predictive performance was used to predict 9-mer CTL epitopes for all the available sets of BoLA alleles using the artificial neural network algorithm. In this server, peptides with binding affinity scores up to 0.5 are considered strong binders [31].

## 2.4. Prediction of Helper T Lymphocyte (HTL) epitopes

Helper T cells, the components of acquired immunity, can stimulate humoral and cellular immune responses in confronting MHC-II endocytosed foreign proteins. Hence, HTL epitopes are noticed in designing immunotherapeutic vaccines. The NetMHCIIpan v4.0 server trained on the advanced artificial neural network method was employed to predict MHC-II binding epitopes for BoLA alleles [32]. HTL epitopes were further evaluated for their ability to induce the IFN-γ cytokine. IFN-γ, as the activator of macrophages, can induce, trigger and modulate innate and adaptive immune responses [33]. For this purpose, all HTL epitopes were submitted to the IFNeitope server using SVM hybrid algorithms [34].

## 2.5. Prediction of Linear B Lymphocyte (LBL) epitopes

B-cell epitopes are mediators to elicit IgG responses in humoral immunity. For 16-mer LBL epitope identification, the sequences were exposed to the ABCpred B-cell epitope prediction server built on the recurrent neural network with a 0.5 threshold value [35]. Besides, BepiPred—2.0 server was employed for LBL epitope prediction with the default parameters [36].

## 2.6. Physicochemical properties evaluation

Different physicochemical properties, including molecular weight (MW), theoretical isoelectric point (pI), instability and aliphatic index, half-life, and grand average of hydropathicity (GRAVY) of the sequences, were predicted by using Expasy Protparam [37].

## 2.7. Secondary structure prediction

For this goal, the chosen amino acid sequences were submitted to the PSIPRED v4.0 server as an accurate secondary structure generator web server with 78.1% accuracy [38]. This server is able to calculate the percentage of the alpha-helix, random coils, and beta-turn in the construct.

## 2.8. 3D structure homology-modelling, refinement, and validation

The GalaxyWEB server was implemented to produce 3D structures of protein constructs by template-based modeling [39]. After that, the models were refined using the GalaxyRefine web server [40]. The overall quality determination was performed via PROCHECK in the SAVES

v6.0 server. Also, the ProSA-web server was used for further structural validation. This server measures the Z-score and figures out the stereochemical quality of structures [41].

## 2.9. Screening of conformational B-cell epitopes

IEDB ElliPro tool was employed to predict conformational or discontinuous B-cell epitopes based on the residue clustering algorithm. The maximum allowed distance and minimum gap between residues were selected as 6.0 Å and 0.5, respectively [42]. Also, epitopic residues were defined through Discotope—2.0 server based on the combination of surface exposure and the log-odds propensity scores [43].

## 2.10. Molecular docking of TLR4 complex and protein constructs

TLR4 complex and protein constructs were docked in ClusPro tools [44]. Owing to the unavailability of bovine TLR4 crystal structure in the protein data bank, and given that the TLR4 gene is conserved in mice and cows, the crystal structure of mouse TLR4 (PDB ID: 7MLM) was used in the docking simulations.

## 3. Results

### 3.1 Protein sequences retrieval and hydrophilicity evaluation

About 12 and 20 amino acid sequences of MAP2191 and FAP-P were retrieved from GenBank in FASTA format. After determining the conserved region of each segment via alignment, the most hydrophilic part of MAP2191 (108 aa length) and FAP-P (103 aa length) were selected based on Kyte and Doolittle Hydrophobicity scale for further analysis [45]. As shown in Fig 1, more residues in both diagrams are in the hydrophilic parts. In the following, these hydrophilic truncated selected segments were named ht-MAP2191 and ht-FAP-P. Blast-p analysis showed no homology of ht-MAP2191 and ht-FAP-P for the bovine host.

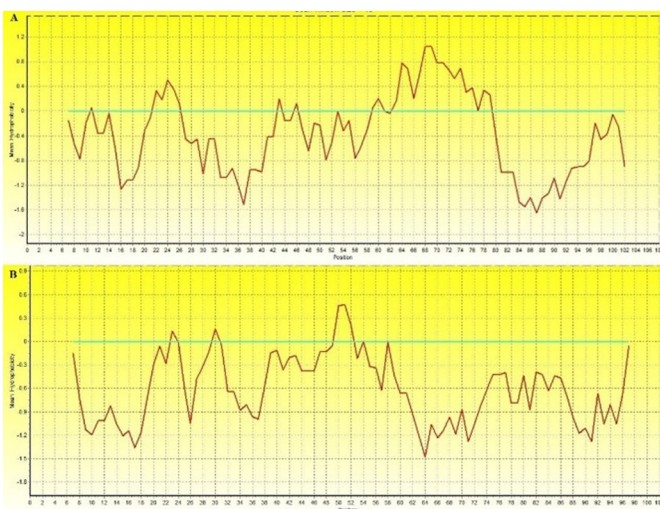

**Fig 1.** Kyte and Doolittle Hydrophobicity plot derived from the MAP2191 (A) and FAP-P (B) conserved amino acid sequences. The hydrophilic residues are below the blue line, and the hydrophobic residues are shown above the blue line.

## 3.2. Immunogenic properties, toxicity, and solubility assessment

ht-MAP2191 and ht-FAP-P showed an antigenic score of 0.4452 and 0.5475, respectively, at a 0.4% threshold and were found to be non-allergenic, non-toxic, and water-soluble. The predicted scale solubility was 0.503 for ht-MAP2191 and 0.638 for ht-FAP-P at a 0.45 threshold with the Protein-Sol server.

## 3.3. CTL and HTL epitopes prediction

ht-MAP2191 and ht-FAP-P sequences were subjected to MHC-I and MHC- II prediction tools to obtain 9-mer and 15-mer lengths T-cell epitopes, respectively (Table 1). All CTL epitopes with the highest prediction score and their corresponding MHC-I alleles are shown in Tables 2 and 3.

In this study, 15-mer HTL epitopes with the IC50 value of ≤50 nM and the highest prediction score against different subtypes of the BoLA DRB3 allele were recognized. Further mining for selecting IFN-γ inducer epitopes illustrated that 32 and 8 HTL epitopes of ht-MAP2191 and ht-FAP-P possess IFN-γ induction characteristics, respectively (S1 Table).

## 3.4. LBL epitopes prediction

Targeted proteins were scrutinized via the ABCPred server, and their continuous B-cell epitopes were ranked in Table 4 according to obtained scores. Eight linear B-cell epitopes were ascertained for each protein, and the maximum prediction score was 0.90.

## 3.5. Physicochemical assessment of ht-MAP2191 and ht-FAP-P

The protein constructs were examined using the Protparam server to verify the physicochemical profile. The results are presented in Table 5.

## 3.6. Secondary structure analysis

The results of the PSIPRED server showed that the secondary structure of ht-MAP2191 consists of 63.9% alpha-helix, 1.9% beta-strand, and 34.2% random coils. The secondary structure of ht-FAP-P includes 13.6% alpha-helix, 20.39% beta-strand, and 66.01% random coils (S1A Fig).

**Table 1. Epitopes prediction profile of ht-MAP2191 and ht-FAP-P for MHC- class I and II.**

|  | ht-MAP2191 | ht-FAP-P |
|---|---|---|
| **MHC-I** |  |  |
| Total no. of epitopes generated | 25 | 18 |
| No. of epitopes binding to one allele | 9 | 2 |
| No. of epitopes binding to ≥2 alleles | 16 | 16 |
| No. of different interacting alleles | 82 | 74 |
| **MHC-II** |  |  |
| Total no. of epitopes generated | 36 | 14 |
| No. of epitopes binding to one allele | 3 | 1 |
| No. of epitopes binding to ≥2 alleles | 33 | 13 |
| No. of IFN-γ inducer epitopes | 32 | 8 |
| No. of different subtypes of interacting DRB3 allele | 37 | 31 |

**Table 2. ht-MAP-2191 CTL epitopes with highest score.**

| Alleles | Peptide | Position |
|---|---|---|
| BoLA-1:01901 | NESKLGPTL | 12 |
| BoLA-1:02901 | NESKLGPTL | 12 |
| BoLA-1:01901 | FEITSGESV | 47 |
| BoLA-1:07401 | NESKLGPTL | 12 |
| BoLA-2:01802 | VPNLAIPEL | 65 |
| BoLA-2:01801 | VPNLAIPEL | 65 |
| BoLA-2:01201 | NSVAAMLEK | 24 |
| BoLA-2:01802 | LPGLKKFEI | 41 |
| BoLA-2:01801 | LPGLKKFEI | 41 |
| BoLA-2:01802 | DPNMPRALF | 88 |
| BoLA-2:01801 | DPNMPRALF | 88 |
| BoLA-2:00501 | VPNLAIPEL | 65 |
| BoLA-2:04401 | KALPGLKKF | 39 |
| BoLA-2:04401 | LIQPFFDYY | 73 |
| BoLA-2:01201 | LSKALPGLK | 37 |
| BoLA-3:06501 | YYNAFVPNL | 60 |
| BoLA-3:03701 | ESVSNGFYY | 53 |
| BoLA-T2c | NLSKALPGL | 36 |
| BoLA-T2c | RLNSVAAML | 22 |
| BoLA-T2a | NSVAAMLEK | 24 |
| BoLA-4:02402 | KALPGLKKF | 39 |
| BoLA-amani.1 | KALPGLKKF | 39 |
| BoLA-T2c | VPNLAIPEL | 65 |
| BoLA-T2c | AIPELIQPF | 69 |
| BoLA-6:01402 | FEITSGESV | 47 |
| BoLA-T2c | LVADNESKL | 8 |
| BoLA-HD6 | RLNSVAAML | 22 |
| BoLA-6:01301 | RLNSVAAML | 22 |
| BoLA-6:01402 | NESKLGPTL | 12 |
| BoLA-T2c | NAFVPNLAI | 62 |
| BoLA-6:03401 | NESKLGPTL | 12 |
| BoLA-5:00301 | RLNSVAAML | 22 |
| BoLA-5:00301 | KLGPTLDRL | 15 |
| BoLA-T2b | NESKLGPTL | 12 |
| BoLA-6:04101 | NESKLGPTL | 12 |
| BoLA-T2a | LSKALPGLK | 37 |

## 3.7. Structural modeling and validations of ht-MAP2191 and ht-FAP-P

The tertiary structure of proteins was obtained from the GalaxyWEB server (Fig 2) and checked for stereochemical quality after refinement. According to the Ramachandran results, the number of amino acids with poor or favored phi/psi angles was investigated (Fig 3). In the best model of ht-MAP219, 97.7% of the residues were in the most favored regions, 1.1% in additional allowed regions, and 1.1% were in generously allowed regions. In ht-FAP-P, these values were 95 and 3.8% for most favored and additional allowed regions, respectively, and 1.2% of residues (TYR 34) were in disallowed regions. Overall quality validation via the ProSA-web server depicted that the Z-score of ht-MAP2191 and ht-FAP-P were -4.26 and -3.98, respectively (Fig 4).

**Table 3. ht-FAP-P CTL epitopes with highest score.**

| Alleles | Peptide | Position |
|---|---|---|
| BoLA-1:02301 | SKVTGPPPM | 3 |
| BoLA-1:01901 | AEANNAKAA | 37 |
| BoLA-2:00801 | ASKPNGQIW | 91 |
| BoLA-2:01201 | GSASYYEVK | 79 |
| BoLA-2:01201 | IVMGRLDQK | 24 |
| BoLA-2:01201 | ASAEANNAK | 35 |
| BoLA-2:00802 | ASKPNGQIW | 91 |
| BoLA-2:01802 | MPDQPPPVA | 11 |
| BoLA-2:01801 | MPDQPPPVA | 11 |
| BoLA-2:02603 | SKVTGPPPM | 3 |
| BoLA-2:02602 | SKVTGPPPM | 3 |
| BoLA-2:02601 | SKVTGPPPM | 3 |
| BoLA-2:01802 | DASKPNGQI | 90 |
| BoLA-2:01801 | DASKPNGQI | 90 |
| BoLA-3:01701 | TGSASYYEV | 78 |
| BoLA-amani.1 | ASKPNGQIW | 91 |
| BoLA-D18.4 | SKVTGPPPM | 3 |
| BoLA-4:02402 | ASKPNGQIW | 91 |
| BoLA-T2a | GSASYYEVK | 79 |
| BoLA-6:01402 | AEANNAKAA | 37 |
| BoLA-T2a | IVMGRLDQK | 24 |
| BoLA-T2a | ASAEANNAK | 35 |
| BoLA-4:02401 | ASKPNGQIW | 91 |
| BoLA-T2c | RINQDSTPL | 63 |
| BoLA-T2c | VANDTRIVM | 18 |

## 3.8. Discontinuous B-cell epitopes prediction

In ht-MAP2191, five discontinuous B-cell epitopes containing amino acid residues ranging from 6 to 24 were found using the ElliPro tool. Also, four conformational B-cell epitopes with a length of 8 to 17 amino acid residues were identified in ht-FAP-P (S1B and S1C Fig). The highest score for epitope prediction in ht-MAP2191 was 0.727; in ht-FAP-P, this score was 0.736. (Table 6). Fig 5 indicates the location of epitopes on the 3D structure of proteins in different colors. The results of the Discotope server showed that 38 and 47 residues of ht-MAP2191 and ht-FAP-P were involved in conformational B-cell epitopes, respectively.

## 3.9. Molecular docking study

The molecular docking was conducted between proteins and TLR4 using the ClusPro server. The complexes with the lowest binding energy of -1023.8 and -877.3 kcal/mol were chosen for ht-MAP2191 and ht-FAP-P, respectively (Fig 6). There were six hydrogen bonds between ht-MAP2191 and TLR4. As well, ht-FAP-P formed eight hydrogen bonds with TLR4 (Fig 7). Table 7 shows the amino acids involved in these hydrogen-bonding interactions.

## 4. Discussion

As a considerable production drop casing by PTB infection, it causes notable welfare and economic implications in the dairy cattle industry worldwide. The lack of treatment options,

**Table 4. LBL epitopes of constructs predicted by ABCPred server.** Common residues detected by both ABCPred and BepiPred servers are marked in blue.

| Name | Rank | Sequence | Start position | Score |
|---|---|---|---|---|
| ht-MAP2191 | 1 | GFRRNDPNMPRALFPW | 84 | 0.90 |
| | 2 | GESVSNGFYYNAFVPN | 53 | 0.88 |
| | 3 | PGLKKFEITSGESVSN | 43 | 0.86 |
| | 4 | LIQPFFDYYFGFRRND | 74 | 0.83 |
| | 5 | VSKQLTGLVADNESKL | 2 | 0.75 |
| | 6 | NESKLGPTLDRLNSVA | 13 | 0.69 |
| | 7 | PNMPRALFPWPHNGIP | 90 | 0.67 |
| | 7 | AAMLEKNRDNLSKALP | 28 | 0.67 |
| ht-FAP-P | 1 | TGPPPMPDQPPPVAND | 7 | 0.90 |
| | 2 | FFMPYPGTRINQDSTP | 56 | 0.87 |
| | 3 | GTRINQDSTPLNGANG | 62 | 0.86 |
| | 4 | PVANDTRIVMGRLDQK | 18 | 0.85 |
| | 5 | QKLYASAEANNAKAAV | 32 | 0.83 |
| | 6 | YYEVKFSDASKPNGQI | 84 | 0.82 |
| | 7 | NGSTGSASYYEVKFSD | 76 | 0.80 |
| | 8 | AVRLGSDMGEFFMPYP | 46 | 0.62 |

**Table 5. Physicochemical characteristics of constructs.**

| Name | Molecular weight kDa | pI | Instability index | Aliphatic index | GRAVY | Estimated half-life |
|---|---|---|---|---|---|---|
| ht-MAP2191 | 12.18 | 9.05 | 39.89 | 76.97 | -0.362 | 30 hours (mammalian reticulocytes, in vitro) >20 hours (yeast, in vivo) >10 hours (Escherichia coli, in vivo) |
| ht-FAP-P | 11.1 | 6.12 | 18.62 | 63.85 | -0.430 | 30 hours (mammalian reticulocytes, in vitro) >20 hours (yeast, in vivo) >10 hours (Escherichia coli, in vivo) |

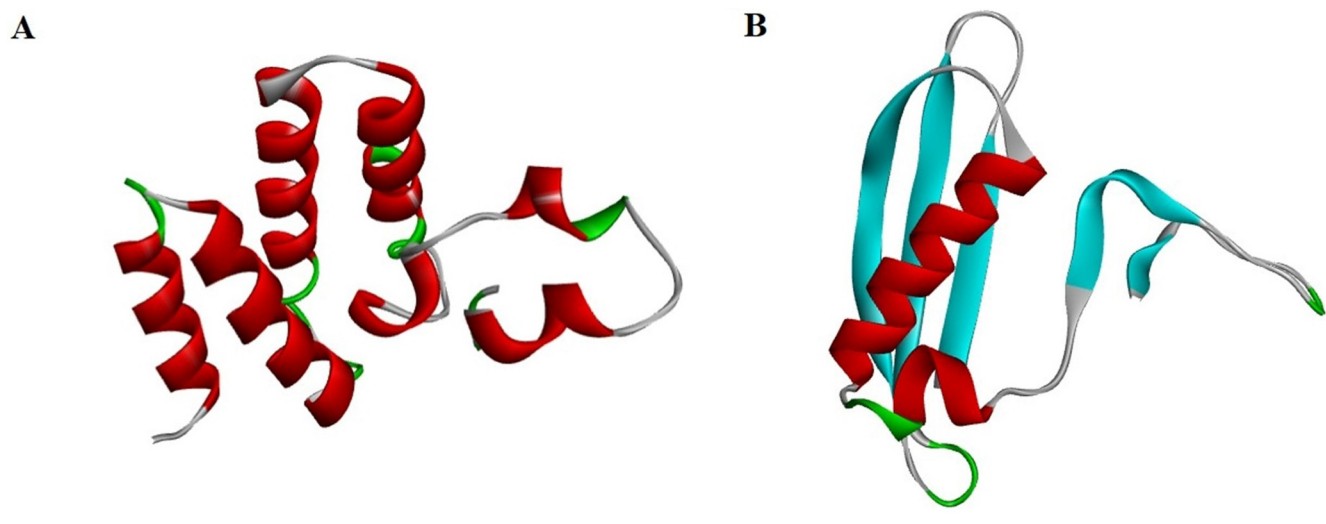

**Fig 2.** The best models' 3D structures of A) ht-MAP2191 and B) ht-FAP-P after refinement.

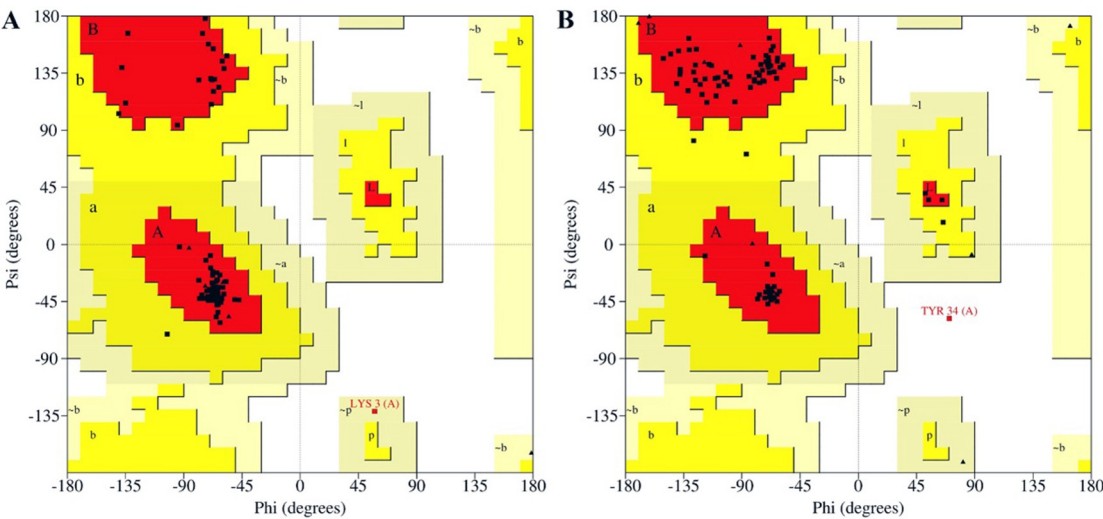

**Fig 3.** Ramachandran plot of A) ht-MAP2191 and B) ht-FAP-P.

including effective vaccines, has led to the emergence of serious obstacles in controlling the disease [46]. Over 146 vaccine studies/trials have been conducted internationally to manage Johne's disease [47]. Choosing an antigen that has the capability to stimulate the immune system comprehensively is the most critical step in vaccine design. Using immunoinformatics tools is a cost-effective approach for selecting potential antigen candidates and has facilitated the time of vaccine development in recent years [48]. In some recent studies, immunoinformatics-based techniques have been employed to detect the epitopes possibly useful for controlling PTB [11, 24].

Considering that to control and clear the infection, the presence of B and T lymphocytes is essential in the early stages of paratuberculosis, a vaccine that can induce both humoral and cellular immune responses and produce a sufficient immunological memory will be successful against this disease [49]. Also, IFNγ is regarded as critical in the Th1-dependent immune

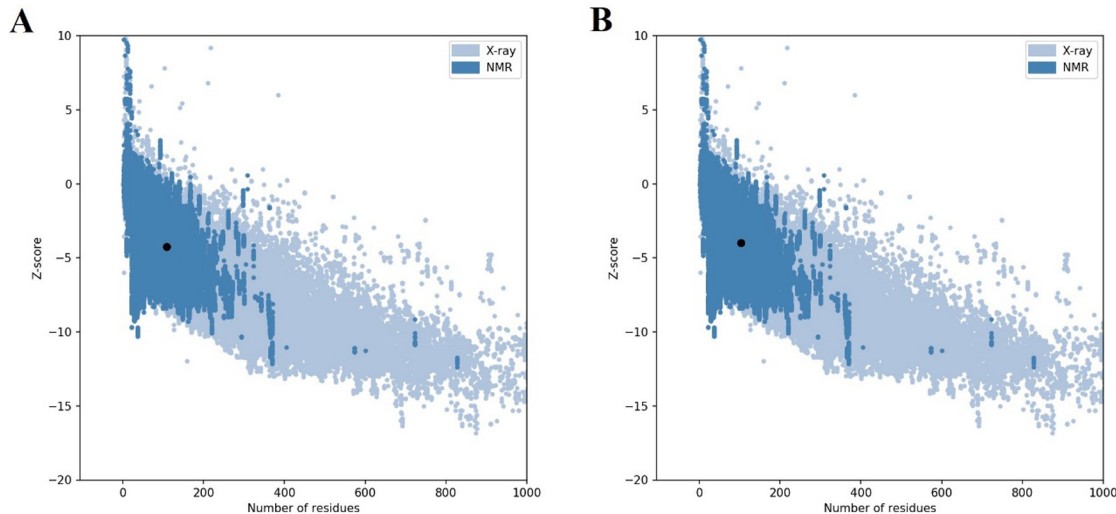

**Fig 4.** Z-score of A) ht-MAP2191 and B) ht-FAP-P.

**Table 6. The amino acid residue content of conformational B-Cell epitopes predicted by the ElliPro server.** Common residues detected by both Ellipro and Discotope servers are marked in blue.

| Name | No. | Residues | Number of residues | Score | Color of epitope in 3D structure |
|---|---|---|---|---|---|
| ht-MAP2191 | 1 | T50, G52, E53, S54, V55, S56, N57, G58, Y61 | 9 | 0.727 | Red |
| | 2 | V1, S2, K3, Q4, L5, T6, G7, L8, A10, D11, E13, S14, K15, L16, G17, P18, T19, R22 | 18 | 0.706 | Green |
| | 3 | R34, D35, N36, L37, S38, K39 | 6 | 0.696 | Yellow |
| | 4 | G83, F84, R85, R86, D88, P89, N90, M91, P92, R93, A94, L95, F96, P97, W98, P99, H100, N101, G102, I103, P104, G105, S107, R108 | 24 | 0.621 | Blue |
| | 5 | L73, I74, Q75 | 3 | 0.574 | Cyan |
| ht-FAP-P | 1 | S3, K4, V5, T6, G7, P8, P9, P10, M11, P12, D13, Q14, P15, P16, P17, V18, A19 | 17 | 0.736 | Red |
| | 2 | N72, G73, A74, N75, G76, S77, T78, I103 | 8 | 0.712 | Green |
| | 3 | M57, P58, Y59, P60, G61, T62, I64, N65, K87, F88, S89, D90, A91, S92, K93, P94, N95 | 17 | 0.672 | Yellow |
| | 4 | G27, R28, L29, D30, Q31, K32, L33, Y34, A35, S36, A37, E38, A39, N40, K43 | 15 | 0.625 | Blue |

responses and leads to the restriction of bacterial multiplication in the early stage of the infection [50, 51].

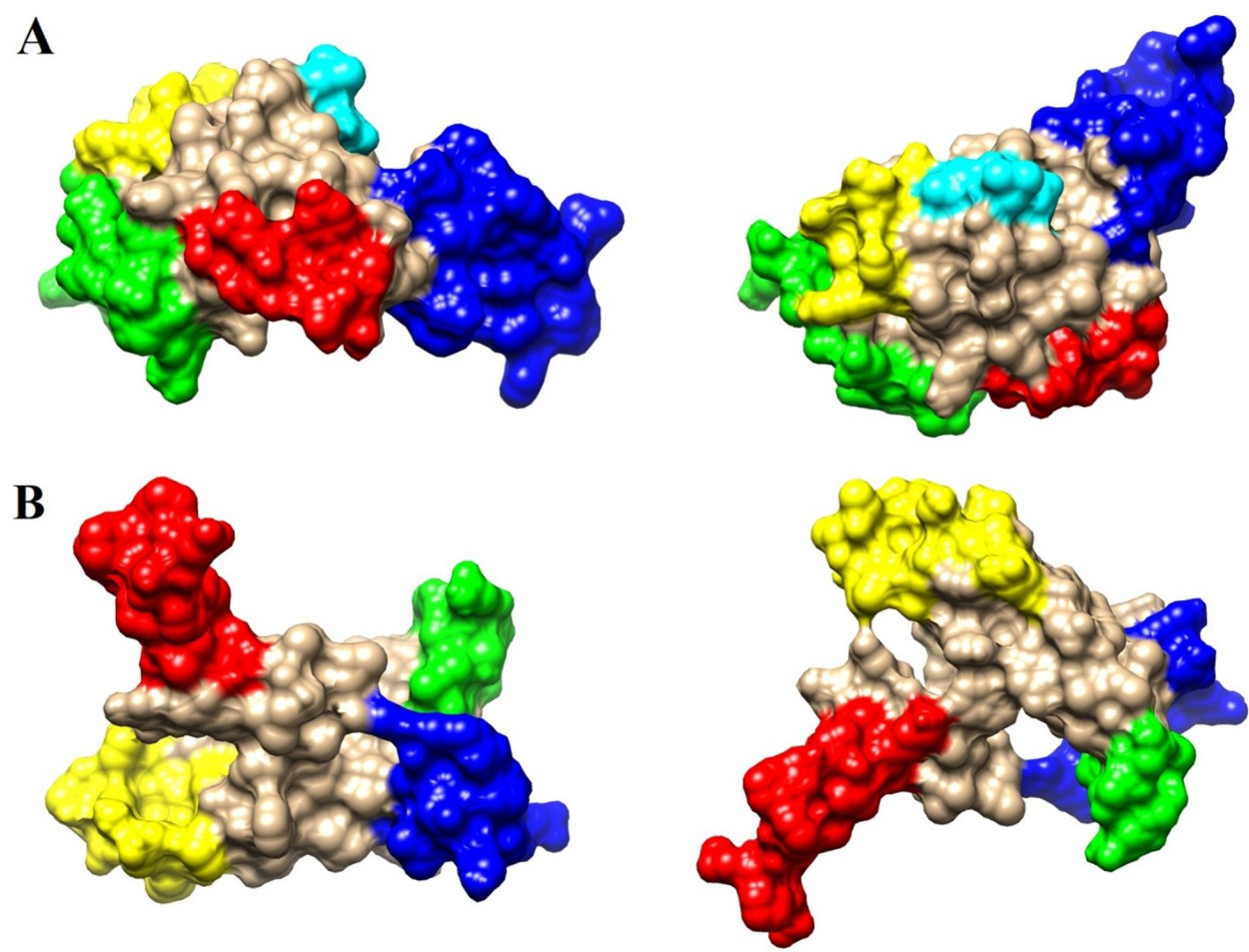

**Fig 5.** Conformational B-cell epitopes of A) ht-MAP2191 and B) ht-FAP-P in two different orientations, distinguished by different colors according to Table 6.

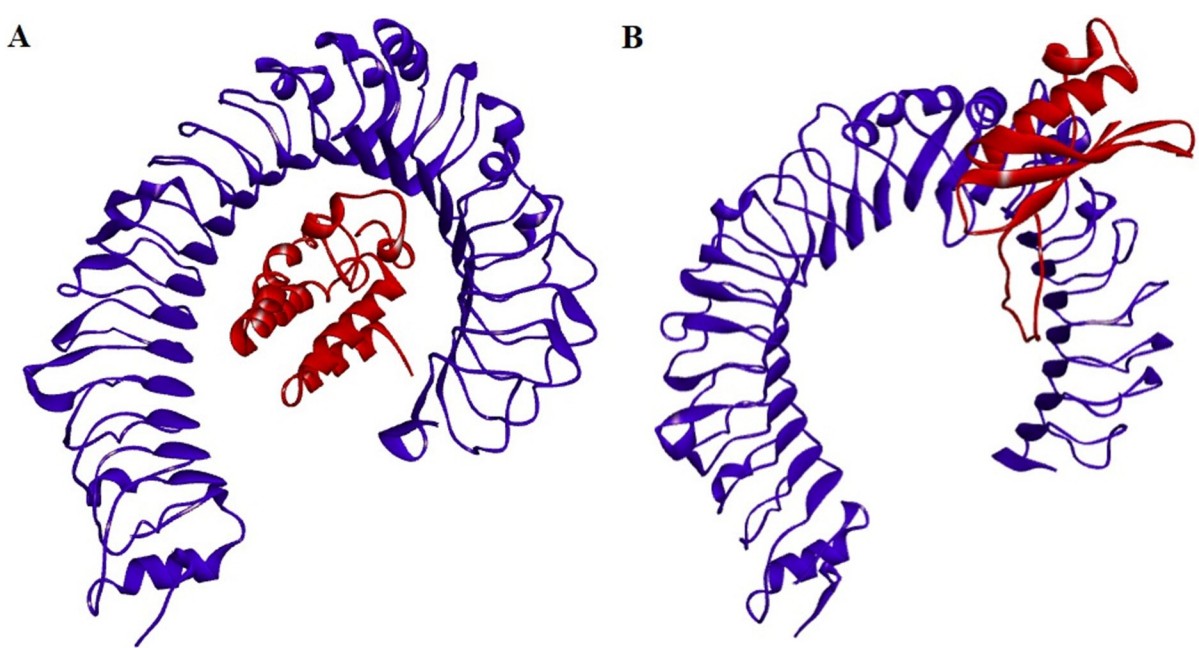

**Fig 6.** Docked complexes of A) ht-Map2191-TLR4 and B) ht-FAP-P-TLR4. The receptor is shown in blue, while ligands are shown in red.

In this work, the potential immunological profile of ht-MAP2191 and ht-FAP-P was explored with the help of immunoinformatics tools to determine their capability as vaccine candidates. In this regard, the initial assessments showed that these selected parts of proteins are non-toxic, non-allergenic, and highly water-soluble, so they had the main characteristics for continuing the evaluations. A comprehensive investigation of epitopes associated with bovine MHC I/II alleles was performed in the next step. A long list of BoLA alleles identified 25 CTL epitopes from ht-MAP2191. As to ht-FAP-P, there were 18 CTL epitopes with strong interactions detectable by BoLA alleles. Among ht-MAP2191 and ht-FAP-P epitopes, KALPGLKKF and ASKPNGQIW could bind to a large number of BoLA molecules, respectively.

BoLA-DRB3 gene, the most polymorphic bovine MHC gene, has shown a predominant role in pathogen presentation to the immune system [52]. After screening of ht-MAP2191 epitopes, 36 different HTL epitopes were found to interact strongly with various subtypes of BoLA DRB3 allele, and YFGFRRNDPNMPRAL was the peptide that could identify by most subtypes. Among 14 HTL epitopes specified in ht-FAP-P, TRIVMGRLDQKLYAS was detected by more BoLA DRB3 molecules. In ht-MAP2191, 16 CTL epitopes overlapped with HTL epitopes, and ht-FAP-P contained six CTL epitopes with binding affinity for MHC class I molecules. Taking into consideration the mentioned outcomes, using ht-MaP2191 and ht-FAP-P as vaccine candidates can provide the overall immunogenicity. Owing to the polymorphism of MHC molecules, promiscuous T cell epitopes, which can bind to different alleles of MHC class I and II, are considered prominent epitopes and are suitable candidates for immunological investigations in MAP [53, 54]. Therefore, it can be said that the results of this study are satisfactory from this point of view.

Because of the IFNγ ability to activate and differentiate immune cells like macrophages and B and T lymphocyte cells, all HTL epitopes were screened to know their IFNγ induction property. The findings of this study showed that most of the HTL epitopes identified in both proteins are qualified in this respect.

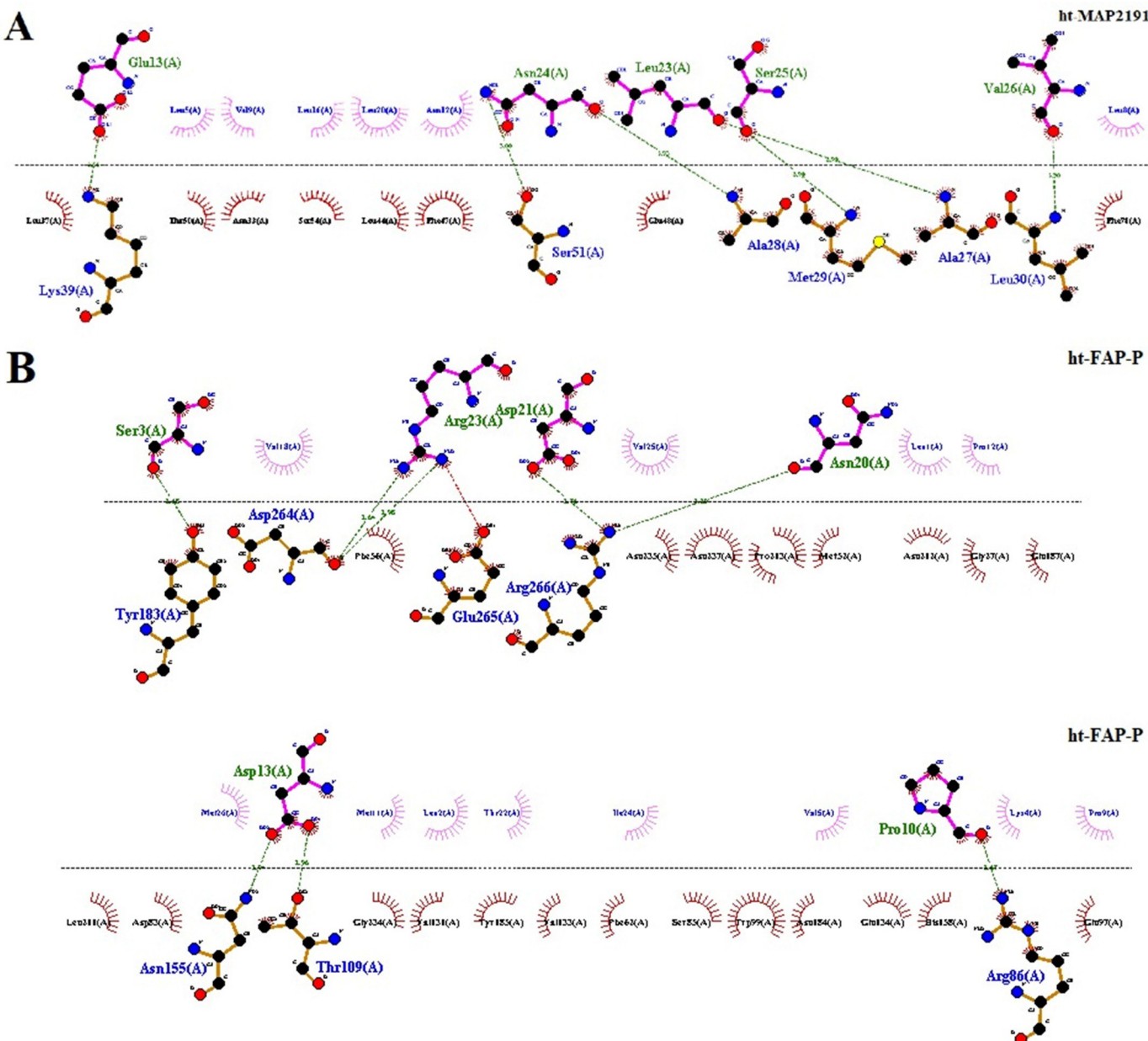

**Fig 7. The map of interactions between ligands and TLR4.** A) ht-Map2191 and B) ht-FAP-P. Green dotted lines represent hydrogen bonds.

The induction of antibody-mediated along with cell-mediated immunity is needed to elicit a prolonged and robust immune response. It has been made clear that the rise of IgG1 levels following vaccination is crucial for preventing MAP infection [55]. Sequential and conformational B-cell epitopes are key mediators in stimulating the humoral immune system responses against the pathogen. Here, the analysis showed that potent linear and conformational B-cell epitopes were involved in the ht-MAP2191 and ht-FAP-P construct.

Understanding of attributes and quality of vaccine antigens is pivotal for estimating the safety and eliciting protective immune responses [56]. The molecular weight of ht-MAP2191 and ht-FAP-P was about 12.18 and 11.1 kDa, respectively, which means they could be an

**Table 7. Amino acids involved in hydrogen-bonding interactions.**

| TLR4 | ht-MAP2191 | Distance (Å) |
|---|---|---|
| Lys39 | Glu13 | 2.56 |
| Ser51 | Asn24 | 3.00 |
| Ala28 | Asn24 | 2.93 |
| Met29 | Ser25 | 2.90 |
| Ala27 | Leu23 | 2.90 |
| Leu30 | Val26 | 2.90 |
| **TLR4** | **ht-FAP-P** | **Distance (Å)** |
| Try183 | Ser3 | 2.68 |
| Asp264 | Arg23 | 2.64 |
| Arg266 | Asn20, Asp21 | 3.20–2.76 |
| Asn155 | Asp13 | 2.94 |
| Thr109 | Asp13 | 2.96 |
| Arg86 | Pro10 | 2.67 |

appropriate choice for large-scale production due to their easy purification method. Proteins with a molecular weight of less than 110 kDa are considered suitable for vaccine design [57]. The results of the theoretical pI value indicated the alkaline nature of ht-MAP2191 and the acidic nature of ht-FAP-P. Their prolonged half-life (30 hours for in vitro and >10 hours for in vivo systems) increases exposure time to the immune system and, as a result, provides the opportunity to stimulate further the immune system. The thermostability of proteins is related to the volume occupied by aliphatic side chains [58]. The value of the aliphatic index of ht-MAP2191 and ht-FAP-P revealed that they could be considered thermostable. The instability index lower than 40 means the protein is stable in biological environments [59]; therefore, ht-MAP2191 and ht-FAP-P were approved in this sense. Their negative grand average of hydropathy (GRAVY) index indicates their hydrophilic characteristic that causes strong interactions with water molecules.

Three-dimensional structures of ht-MAP2191 and ht-FAP-P were built and subjected to the refinement process to improve the quality. After that, the data obtained from the Ramachandran plot and ProSA-web server approved the structures of refined models.

One of the necessities of creating an effective immune response is establishing an interaction between the vaccine and the immune cell. As reported by (Lee et al., 2009), FAP can modulate adaptive immune responses to *M. avium* subsp. *paratuberculosis* by activation of dendritic cells through TLR4 [20]. Besides, there is evidence that TLR4 polymorphism is involved in susceptibility to MAP infection [60]. On the other hand, it has been revealed that TLR4 contributes to *Mycobacterium tuberculosis* infection eradication [61]. So, the molecular docking simulation was carried out for TLR4 as a receptor and ht-MAP2191 and ht-FAP-P as ligands. The results of the docking process showed the strong interactions of vaccine candidates against TLR4.

## 5. Conclusion

The findings of the present study demonstrate that ht-MAP2191 and ht-FAP-P possess potent epitopes that confer the ability to stimulate adaptive immunity and IFNγ induction property against MAP infection. Also, it is expected that they can induce a pro-inflammatory response following the activation of TLR4. It is worth mentioning that further in vitro and in vivo experiments are underway in the second phase of this study.

## Supporting information

**S1 Table. IFN-γ inducer HTL epitopes.**
(DOCX)

**S1 Fig.**
(DOCX)

## Author Contributions

**Conceptualization:** Maryam Sadat Moezzi, Abdollah Derakhshandeh, Farhid Hemmatzadeh.

**Formal analysis:** Maryam Sadat Moezzi, Abdollah Derakhshandeh, Farhid Hemmatzadeh.

**Funding acquisition:** Abdollah Derakhshandeh.

**Methodology:** Maryam Sadat Moezzi, Abdollah Derakhshandeh, Farhid Hemmatzadeh.

**Software:** Maryam Sadat Moezzi, Abdollah Derakhshandeh.

**Supervision:** Abdollah Derakhshandeh, Farhid Hemmatzadeh.

**Validation:** Maryam Sadat Moezzi.

**Writing – original draft:** Maryam Sadat Moezzi, Abdollah Derakhshandeh, Farhid Hemmatzadeh.

**Writing – review & editing:** Maryam Sadat Moezzi, Abdollah Derakhshandeh, Farhid Hemmatzadeh.

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
