## [Decision Letter · Decision Letter 0]

4 Oct 2022

PONE-D-22-24057Immunoinformatics Analysis of Candidate Proteins for Controlling Bovine ParatuberculosisPLOS ONE

Dear Dr. Derakhshandeh,

Thank you for submitting your manuscript to PLOS ONE. After careful consideration, we feel that it has merit but does not fully meet PLOS ONE’s publication criteria as it currently stands. Therefore, we invite you to submit a revised version of the manuscript that addresses the points raised during the review process. Your manuscript has been reviewed  and a minor revision is needed.  

We look forward to receiving your revised manuscript.

Kind regards,

Yung-Fu Chang

Academic Editor

PLOS ONE

Journal Requirements:

"This work was financially supported partially by a grant from Shiraz University (98GCU2M163973) and also Iran National Science Foundation (INSF) (No. 98023138). "

"This work was financially supported partially by a grant from Shiraz University (98GCU2M163973) and also Iran National Science Foundation (INSF) (No. 98023138). "

"This work was financially supported partially by a grant from Shiraz University (98GCU2M163973) and also Iran National Science Foundation (INSF) (No. 98023138). "

Reviewers' comments:

Reviewer's Responses to Questions

**Comments to the Author**

1. Is the manuscript technically sound, and do the data support the conclusions?

Reviewer #1: Yes

2. Has the statistical analysis been performed appropriately and rigorously? 

Reviewer #1: N/A

3. Have the authors made all data underlying the findings in their manuscript fully available?

Reviewer #1: Yes

4. Is the manuscript presented in an intelligible fashion and written in standard English?

Reviewer #1: Yes

5. Review Comments to the Author

Reviewer #1: The study performed on the available data on NCBI and other database. The authors have performed all required analysis for the prediction of protective vaccine against MAP infection. Thay have considered all parameters to be present in a protective antigen candidate. The current manuscript does not have any statistical analysis. All data are available without any restriction and language of authors are technically sound and easy to understand.

6. PLOS authors have the option to publish the peer review history of their article (what does this mean?). If published, this will include your full peer review and any attached files.

Reviewer #1: No

---

## [Author Response · Author response to Decision Letter 0]

18 Oct 2022

Editor in Chief,

PLOS ONE, 

Dear Sir/Madam

We really appreciate reviewer comments which helped us to revise and improve the manuscript. Most comments seemed to be fair and reasonable, so we revise the manuscript according to their advice and suggestions. We believe that the contents and the clarity of our paper are improved in the re-submitted version. Below are point-by-point responses to the reviewers and the journal requirements comments. The revised manuscript was submitted as ‘Highlighted Revised Manuscript’ file which all the requested corrections was addressed in highlighted in yellow. The second file as a separate file labeled 'Manuscript' was also uploaded. I hope the responses satisfy you and the editorial groups in order to publish our paper.

Sincerely Yours,

Abdollah Derakhshandeh

 Department of Pathobiology, School of Veterinary Medicine, Shiraz University, Shiraz, 71345-1731, Iran. Tel: +98-7136138666, Mobile: +98-9171825358, Fax: +98-7132286940, Email: drkhshnd77@gmail.com. 

Journal Requirements:

Response: Yes, All style requirements was done.

"This work was financially supported partially by a grant from Shiraz University (98GCU2M163973) and also Iran National Science Foundation (INSF) (No. 98023138). "

Response: The sentence was added in funding statement and highlighted in yellow

"This work was financially supported partially by a grant from Shiraz University (98GCU2M163973) and also Iran National Science Foundation (INSF) (No. 98023138). "

Please remove any funding-related text from the manuscript and let us know how you would like to update your Funding Statement. Currently, your Funding Statement reads as follows: "This work was financially supported partially by a grant from Shiraz University (98GCU2M163973) and also Iran National Science Foundation (INSF) (No. 98023138). "

Response: Please include the statement "This work was financially supported partially by a grant from Shiraz University (98GCU2M163973) and also Iran National Science Foundation (INSF) (No. 98023138)." in you online submission on behalf of us. We deleted funding information in the Acknowledgments section or other areas of our manuscript.

4. Please review your reference list to ensure that it is complete and correct

Response: All references are complete and correct.

Reviewer Comments:

I have reviewed the manuscript entitled "Immunoinformatics Analysis of Candidate Proteins for Controlling Bovine Paratuberculosis" by Moezzi et. al., The authors have evaluated the efficacy of two proteins MAP2191 and FAP-P to be a potent vaccine candidate against MAP infection using immunoinformatics approaches. 

Response: Thanks for kind words and sentences about our study. 

1. The authors have selected only one proteins MAP2191 out of cluster of 6 genes, why only one?, it has been mentioned that all these proteins are involved MAP invasion, survival and virulence (Hemati Z, 2018; Semret M, 2005). 

Response:

Thank you for the comment. Because of identifying and submitting the nucleotide sequence of the MAP2191 gene and some experimental works have been performed on the MAP2191 protein in our lab. So, in this study, the authors have decided to consider the MAP2191 amino acid sequence for immunoinformatics evaluations. Finally, when any researchers work on MAP, she/he must consider the cross reactivity with Mycobacterium bovis. MAP 2191shows no cross reactivity with TB. 

2. The author did not mention anywhere whether these protein will act as protective antigen against all serovars or single serovars of MAP.

Response:

Thanks for raised comments. As stated at the beginning of the results, conserved regions of these two proteins from 12 and 20 amino acid sequences of MAP2191 and FAP-P were selected, respectively. Due to the consideration of conserved regions of different sequences available in the GenBank, the authors believe these two proteins can act as protective antigens against different serovars of MAP.

3. The author should also explore the Virulence database.

Response:

Many thanks for your suggestion. Regarding the comment of the respected reviewer, the authors have searched the virulence database (VFDB) as the reference database. Two genomes are available for MAP in this database (k10 and MAP4), and their sequences have been released in NCBI. So, the amino acid sequences of MAP2191 and FAP-P that are available in NCBI have been considered in this study.

4. The author can explore the confident B-Cell epitope by using two or more different tool and common prediction by all these will be considered confident B-cell epitope (conformational as well as linear epitope).

Response:

Many thanks for your good comment. To comply with the reviewer's comment, two servers were considered for linear and conformational epitope prediction in the revised version of the manuscript. The results are given in Table 4 and Table 6, respectively.

---

## [Decision Letter · Decision Letter 1]

3 Nov 2022

Immunoinformatics Analysis of Candidate Proteins for Controlling Bovine Paratuberculosis

PONE-D-22-24057R1

Dear Dr. Derakhshandeh,

We’re pleased to inform you that your manuscript has been judged scientifically suitable for publication and will be formally accepted for publication once it meets all outstanding technical requirements.

Kind regards,

Yung-Fu Chang

Academic Editor

PLOS ONE

Additional Editor Comments (optional):

Reviewers' comments:

Reviewer's Responses to Questions

**Comments to the Author**

1. If the authors have adequately addressed your comments raised in a previous round of review and you feel that this manuscript is now acceptable for publication, you may indicate that here to bypass the “Comments to the Author” section, enter your conflict of interest statement in the “Confidential to Editor” section, and submit your "Accept" recommendation.

Reviewer #1: All comments have been addressed

Reviewer #2: (No Response)

2. Is the manuscript technically sound, and do the data support the conclusions?

Reviewer #1: Yes

Reviewer #2: Yes

3. Has the statistical analysis been performed appropriately and rigorously? 

Reviewer #1: N/A

Reviewer #2: N/A

4. Have the authors made all data underlying the findings in their manuscript fully available?

Reviewer #1: Yes

Reviewer #2: Yes

5. Is the manuscript presented in an intelligible fashion and written in standard English?

Reviewer #1: Yes

Reviewer #2: Yes

6. Review Comments to the Author

Reviewer #1: All the 4 comments raised by me in earlier version of paper gas been properly addressed by the authors. They also performed the required analysis for predicting the Confident B-cell epitopes.

Reviewer #2: The authors have provided a detailed study to test the in vitro efficacy of vaccine candidates. The authors have provided sufficient data to support the informatics analysis. In addition, authors have addressed all the reviewers comments.

7. PLOS authors have the option to publish the peer review history of their article (what does this mean?). If published, this will include your full peer review and any attached files.

Reviewer #1: **Yes: **Dr Mohd Abdullah

Reviewer #2: No

---

## [Editor Report · Acceptance letter]

10 Nov 2022

PONE-D-22-24057R1 

Immunoinformatics Analysis of Candidate Proteins for Controlling Bovine Paratuberculosis 

Dear Dr. Derakhshandeh:

I'm pleased to inform you that your manuscript has been deemed suitable for publication in PLOS ONE. Congratulations! Your manuscript is now with our production department. 

Kind regards, 

on behalf of

Dr. Yung-Fu Chang 

Academic Editor

PLOS ONE